# Obstetric Outcomes after Perforation of Uterine Cavity

**DOI:** 10.3390/jcm11154439

**Published:** 2022-07-30

**Authors:** Polina Schwarzman, Yael Baumfeld, Salvatore Andrea Mastrolia, Shimrit Yaniv-Salem, Elad Leron, Tali Silberstein

**Affiliations:** 1Department of Obstetrics and Gynecology, Soroka University Medical Center, Ben-Gurion University of the Negev, Beer Sheva 8410101, Israel; schwarzmanp@gmail.com (P.S.); yaelkup@yahoo.com (Y.B.); yanivshi@bgu.ac.il (S.Y.-S.); leron@bgu.ac.il (E.L.); talisil@bgu.ac.il (T.S.); 2Clinical Research Center, Faculty of Health Sciences, Soroka University Medical Center, Ben-Gurion University of the Negev, Beer Sheva 8410101, Israel; 3Department of Obstetrics and Gynecology, Ospedale Civile Umberto I, Via Ruvo, 108, 70033 Corato, Italy

**Keywords:** uterine perforation, gynecological procedure, obstetric outcomes, uterine rupture

## Abstract

We aimed to evaluate the pregnancy characteristics and obstetric outcomes in patients after perforation of the uterus. **Study design:** A retrospective cohort study was conducted and included all patients who were diagnosed with uterine perforation and treated in a tertiary referral medical center between the years 1996 and 2018. Up to two deliveries after perforations were investigated. **Results:** During the study period, 51 women were diagnosed with uterine perforation during gynecological procedures, including intrauterine device (IUD) insertion. The mean age of patients at the time of diagnosis was 27.9 (±4.7) years. The majority, 76.5% (*n* = 39), experienced perforation during IUD insertion, and 23.5% (*n* = 12) of the patients experienced perforation during surgical procedures. Most of the patients were multiparous or grand multiparous, 45.8. % (*n* = 22) and 39.6% (*n* = 19) respectively. Anteflexed uterus was found in 86.4% of the patients (*n* = 38). Five patients (9.8%) had pelvic abscesses after the IUD insertion. A total of 50 patients had 71 deliveries subsequent to uterine perforation. One patient experienced intrauterine fetal death due to fetal malformations. One patient experienced uterine rupture. No other major obstetric complications were noted. **Conclusions:** Uterine perforation may be associated with adverse obstetric outcomes. The possibility of uterine rupture must be considered while managing the deliveries of patients after uterine perforation. Moreover, a larger cohort and further studies are needed to establish an association between uterine perforation and adverse outcomes of the subsequent deliveries.

## 1. Introduction

Perforation of the uterus may be a complication that potentially results from any kind of uterine manipulation [1,2]. The incidence of perforations varies from 0.1 to 5%, depending on the procedure and the performer’s skill level [3,4,5,6]. While these numbers are relatively small, it is thought that the actual prevalence of perforations is much higher, as many perforations are unrecognized or underreported [7]. Typically, the damage occurs during dilatation of the cervix or the introduction of an operative instrument [8]. Common locations for uterine perforation are the uterine fundus, the uterine anterior wall and the cervix [6,9]. Different risk factors have been identified for perforation during uterine procedures, including the following: stenotic or scarred cervix (primigravida, cervix after previous procedure or conization); altered position and direction of the uterus (retroflection, hyperanteflection, deformity after cesarean section, fibroids or other uterine pathology); and reduced strength of the myometrium (pregnancy, multiparity, infection, postpartum period and lactation, especially for IUD insertion) [4,8,9,10]. Patient outcomes after uterine perforation are usually good, unless the complication was diagnosed late or there was intraabdominal organ damage [3]. The question arises, however, about the obstetric outcomes in those patients. There are several case reports that describe uterine rupture after perforation [11,12]. Some authors hypothesize that cases of uterine rupture in an un-scarred uterus are due to undiagnosed perforation and this hypothesis is supported by the fact that about 50% of patients with uterine rupture had previous surgical intervention [13].

The aim of our study is to evaluate obstetrical outcomes following uterine perforation.

## 2. Materials and Methods

A retrospective, cohort study was conducted at Soroka University Medical Center (SUMC) for patients treated between the years 1996 and 2018. We included all patients with a confirmed diagnosis of uterine perforation, who were treated in our hospital and had subsequent deliveries. Data including demographic characteristics, general health status, perforation management details and surgical reports were collected from the patient’s electronic medical records. Pregnancy, delivery characteristics and perinatal outcomes were gathered from the computerized obstetric database of the Obstetrics and Gynecology department. Up to two deliveries after perforations were included. Patients with missing data were excluded from the analysis. Informed consent was not obtained due to the retrospective study design. It was waived by the Institutional Review Board of Soroka University Medical Center (#SOR-0149-17 approved on 3 August 2017).

Statistical analysis was performed with the SPSS package, version 20 (SPSS Inc, Chicago, IL, USA). Categorical variable data were presented using percentiles and statistical significance was tested using the X^2^ and Fisher’s exact test, as appropriate. Continuous variable data were presented using mean and standard deviation and Student’s t-test was used for statistical analysis.

The Institutional Review Board of Soroka University Medical Center approved the study that was performed in accordance with the ethical standards laid down in the 1964 Declaration of Helsinki and its later amendments (#SOR-0149-17 approved on 3 August 2017). The study was designed according to the STROBE [14] statement checklist with items for cohort studies.

## 3. Results

During the study period, 51 women were identified with a diagnosis of uterine perforation and subsequent delivery. The mean age of patients at the time of diagnosis was 27.9 ± 4.7 years. The majority of the patients were multiparous or grand multiparous, 45.8% (*n* = 22) and 39.6% (*n* = 19), respectively. The demographic characteristics are presented in Table 1. The most common procedure to cause the perforation was intrauterine device (IUD) insertion in outpatient clinics, which accounted for 76.5% of the patients (*n* = 39). The rest, 23.5% (*n* = 12), of the patients experienced perforation during surgical procedures, predominantly during dilatation and curettage (Table 2). Anteflexed uterus was found in 86.4% of the patients (*n* = 38). The most frequent location of damage was the parametria (*n* = 17, 34%), probably after cervix or uterine isthmus perforation. Five patients (9.8%), who were referred for laparoscopy due to lost IUD, were diagnosed with capsulated pelvic abscesses. The abscess was asymptomatic or caused mild chronic abdominal pain; laparoscopy in all five patients revealed adhesions of the omentum and bowel around the region of IUD location in the abdomen. Background and uterine condition characteristics are presented in the Table 3. The study group of 51 patients had 71 deliveries subsequent to the uterine perforation. The median of time from perforation to first delivery was 36.50 (18.83–63.77) months. One patient experienced intrauterine fetal death due to fetal malformations. One patient experienced uterine rupture at 24 weeks gestation, following fundal-posterior wall perforation in her previous pregnancy. The perforation occurred during postpartum curettage due to adherent placenta. Subsequent uterine rupture manifested with acute and significant abdominal pain at 24 weeks of gestation and an urgent cesarean section was performed, during which a uterine defect was identified in the perforation region and sutured. This pregnancy resulted in neonatal death at day 3, due to prematurity complications. No other major complications were associated with any of the pregnancies. Most of patients had vaginal delivery, 84.5% (*n* = 60), with a mean gestational age of 38.29 ± 2.9 weeks. Placenta previa was diagnosed in one case, abruption of placenta in two cases, three patients had post-partum hemorrhage (PPH), and four patients underwent manual removal of the placenta (manualysis). Pregnancy course and obstetric outcomes are presented in the Table 4.

## 4. Discussion

We conducted a retrospective cohort study in a tertiary referral medical center. We identified 51 patients who had deliveries following uterine perforation. Most of our patients were multiparous, a known risk factor for perforation [15]. The majority of our study population had an anteflexed uterus, as opposed to previous publications that described a retroverted uterus to be a significant risk factor for perforation [16]. A possible explanation may be uterus hyperanteflexion in these patients, but unfortunately, these data were not available. The prevalence of post-partum hemorrhage among our population correlated with that reported in previous studies [17].

We found an increased number of pathological placental conditions in our study group. Manualysis after delivery was documented in four patients (5.6%), while adherent placenta complicates 1–3% of deliveries in the general population [18] and the reported incidence of the manual removal of the placenta is 2.7% [19]. Placenta previa was also found in 1.4% of our patients, while the incidence in the general population is reported to be 0.3–0.5% [20]. 

Uterine rupture occurred in one patient, manifesting with abdominal pain in a nonlaboring patient. Uterine rupture is a dramatic and rare complication, which mostly occurs after cesarean section. The incidence of uterine rupture after cesarean delivery is 5.3/10,000 births [21,22], whereas the incidence of uterine rupture in an unscarred uterus is estimated as 0.6/10,000 [23]. Higher rates of maternal and fetal mortality were found in cases with the rupture of an unscarred uterus, possibly as a consequence of this complication being unexpected [24]. Unrecognized uterine perforation from a previous uterine procedure may be the risk factor for uterine rupture in those cases [11,12,13,25]. In addition, a potential explanation for uterine rupture after previous perforation may be associated with abnormal and not well-organized uterine activity, due to the interruption of the circuit of normal muscular fibers.

Our study has several limitations. This is a retrospective cohort study, limited by its sample size. The rare nature of the condition, specifically during the years of fertility, the underdiagnoses and underreporting all account for our small sample size. We also have heterogeneity in perforation type and treatment. Our data suggest that previous uterine perforation may be followed by obstetric complications, although we cannot establish a significant correlation, due to the limitations of our study. This supports the implementation of preventive measures during uterine procedures, such as cervical priming for all patients and the use of sonographic guidance in such procedures, when appropriate [26].

Our findings emphasize the importance of previous history of uterine manipulation or perforations in the management of a current pregnancy and further studies are needed to establish appropriate recommendations.

## 5. Conclusions

Precautions should be taken during all intrauterine procedures, especially in multiparous women. Ultrasound guidance may be considered, according to the circumstances. 

Uterine perforation may be associated with adverse obstetric outcomes. The possibility of uterine rupture must be considered while managing the deliveries of patients after uterine perforation. Moreover, a larger cohort and further studies are needed to establish an association between uterine perforation and adverse outcomes of subsequent deliveries.

## Figures and Tables

**Table 1 jcm-11-04439-t001:** Demographic characteristics for patients delivered following perforation of uterus.

Variables	Value
Ethnicity	
Jewish	19 (26)
Bedouin	37 (74)
Maternal age	27.94 ± 4.74
Gravidity	4 (2–6)
Parity	4 (2–6)
Primiparity	7 (14.6)
Grand multiparity	19 (39.6)

Data are presented as number (percentage), mean ± standard deviation and median (interquartile range).

**Table 2 jcm-11-04439-t002:** Perforation characteristics and management data for patients who delivered following perforation of uterus.

Variables	Value
Perforation site	
Fundus	9 (18)
Posterior wall	12 (24)
Anterior wall	7 (14)
False root	5 (10)
Parametria (through cervix)	17 (34)
Perforation event	
IUD insertion	39 (76.5)
Surgical procedure	12 (23.5)
Management	
Follow up	7 (13.8)
Laparoscopic IUD removal	41 (80.3)
Laparoscopic coagulation or suture	3 (5.9)
Abscess and adhesions	5 (10.0)
Time from perforation to delivery (months)	36.50 (18.83–63.77)

Data are presented as number (percentage), mean ± standard deviation and median (interquartile range). IUD, intrauterine device.

**Table 3 jcm-11-04439-t003:** Background and uterine condition characteristics for patients who delivered following perforation of uterus.

Variables	Value
Background medical conditions	
Diabetes mellitus	1 (2)
Hypothyroidism	2 (3.9)
Obesity	6 (11.8)
Recurrent abortions	3 (5.9)
Uterine conditions	
Previous cesarean sections	8 (11.7)
Septate uterus	1 (2)
Bicornuate uterus	1 (2)
Uterus myomatosus	1 (2)
Uterine position	
Anteversion	38 (86.4)
Retroversion	12 (13.6)

Data are presented as number (percentage).

**Table 4 jcm-11-04439-t004:** Pregnancy complications and obstetric outcomes among patients following perforation of uterus.

Variables	Value
Preterm contractions	3 (4.2)
PROM	2 (2.8)
Gestational diabetes mellitus	2 (2.8)
Preeclampsia	1 (1.4)
Placenta previa	1 (1.4)
Placenta accreta	0 (0.0)
Placental abruption	2 (2.8)
Gestational age at delivery	38.29 ± 2.89
Spontaneous vaginal delivery	60 (84.5)
Vacuum delivery	1 (1.4)
Cesarean section	10 (14.1)
Manual removal of the placenta	4 (5.6)
Postpartum hemorrhage	3 (4.2)
Uterine rupture	1 (1.4)
Birthweight	3270.9 ± 721.2
Apgar at 1′	9 (9–9)
Apgar at 5′	10 (10–10)

Data are presented as number (percentage), mean ± standard deviation and median (interquartile range). PROM, premature rupture of membranes.

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
