# Peer review of "Obstetric Outcomes after Perforation of Uterine Cavity"

_jcm, 2022, doi:10.3390/jcm11154439_

Round 1

Reviewer 1 Report

Nice work. Always interesting theme as uterine rupture is life threateningh situation. My presumption in research would be about scaring uterine muscle that can be cause of dissinergic uterine contractions potentialy leading to uterine rupture. 

Author tried to explain association between iatrogenic injury (performation) of uterus and uterine rupture during delivery. I would put accent in discussion on dissinergic contraction that could evolve as uterine wall is damaged in previous procedures. Only fibrosis is not adequate explanation for rupture as we know that all ruptures are on healthy tissue not on damaged. Fibrosis-dissinergic contractions-uterine rupture. 

Author Response

Reviewer #1

Comment #1: Nice work. Always interesting theme as uterine rupture is life threateningh situation. My presumption in research would be about scaring uterine muscle that can be cause of dissinergic uterine contractions potentialy leading to uterine rupture.

Response to comment #1: We thank the Reviewer for his complimenting comment and for the interesting insight regarding dissinergic uterine contractions, that was added to the Discussion section of the manuscript.

Comment #2: Author tried to explain association between iatrogenic injury (performation) of uterus and uterine rupture during delivery. I would put accent in discussion on dissinergic contraction that could evolve as uterine wall is damaged in previous procedures. Only fibrosis is not adequate explanation for rupture as we know that all ruptures are on healthy tissue not on damaged. Fibrosis-dissinergic contractions-uterine rupture.

Response to comment #2: As explained in the response to comment #1 we found the Reviewer’s suggestion interesting and added a paragraph to the Discussion section, on page #10 of the revised version of the manuscript.

Reviewer 2 Report

I thank you for the possibility of reviewing this manuscript. I believe that the argument is interesting and that the manuscript deserves some merits; however, before considering it for publication, major revisions are required.
Here are my points:
1) The authors stated this to be "A retrospective, case-control study," but it is not a case-control. In the paper, no controls are presented. However, the study can be classified as a case series or cohort, depending on how the recruitment was done. Please amend this issue throughout the manuscript.
2) In Tables 1, 2, and 4, some continuous variables are presented as mean (±standard deviation). Hower these variables seem to be non-parametric and require to be presented as the median and interquartile range (e.g., median (1st quartile-3rd quartile) of the distribution).
3) The methods section should comply with STROBE guidelines [von Elm E, Altman DG, Egger M, Pocock SJ, Gøtzsche PC, Vandenbroucke JP; STROBE Initiative. The Strengthening the Reporting of Observational Studies in Epidemiology (STROBE) statement: guidelines for reporting observational studies. Ann Intern Med. 2007 Oct 16;147(8):573-7. doi: 10.7326/0003-4819-147-8-200710160-00010. Erratum in: Ann Intern Med. 2008 Jan 15;148(2):168. PMID: 17938396.]
4) The discussions section should also be re-written considering all the points raised and the STROBE guidelines. Moreover, the statement, "Our data suggests that previous uterine perforation may be a significant risk factor for obstetric complications, " is absolutely not supported by the data of this study and not only because it is underpowered. In addition, the conclusion "Uterine perforation may impair obstetric outcomes" is a clear overstatement. Please amend.

Author Response

Reviewer #2

Comment #1: The authors stated this to be "A retrospective, case-control study," but it is not a case-control. In the paper, no controls are presented. However, the study can be classified as a case series or cohort, depending on how the recruitment was done. Please amend this issue throughout the manuscript.

Response to comment #1: The Reviewer is right, the type of the study was corrected as suggestet. 

Comment #2: In Tables 1, 2, and 4, some continuous variables are presented as mean (±standard deviation). However, these variables seem to be non-parametric and require to be presented as the median and interquartile range (e.g., median (1st quartile-3rd quartile) of the distribution).

Response to comment #2: We thank the Reviewer for his comment. Data have been amended as suggested.

Comment #3: The methods section should comply with STROBE guidelines [von Elm E, Altman DG, Egger M, Pocock SJ, Gøtzsche PC, Vandenbroucke JP; STROBE Initiative. The Strengthening the Reporting of Observational Studies in Epidemiology (STROBE) statement: guidelines for reporting observational studies. Ann Intern Med. 2007 Oct 16;147(8):573-7. doi: 10.7326/0003-4819-147-8-200710160-00010. Erratum in: Ann Intern Med. 2008 Jan 15;148(2):168. PMID: 17938396.]

Response to comment #3: We thank the Reviewer for his comment. We have used the Strobe statement checklist that was applied to the Methods section.

Comment #4: The discussions section should also be re-written considering all the points raised and the STROBE guidelines.

Response to comment #4: We thank the Reviewer for his comment and applied the Strobe statement guideline and checklist to the Discussion as done with the Methods section.

Comment #5: Moreover, the statement, "Our data suggests that previous uterine perforation may be a significant risk factor for obstetric complications, " is absolutely not supported by the data of this study and not only because it is underpowered.

Response to comment #5: The Reviewer is right. Therefore, according to his comment, we modified the Discussion section that now reads “Our data suggest that a previous uterine perforation may be followed by obstetric complications, although we cannot establish a significant correlation due to the limitations of our study”.

Comment #6: In addition, the conclusion "Uterine perforation may impair obstetric outcomes" is a clear overstatement. Please amend.

Response to comment #6: We thank the Reviewer for his comment. Our cohort was too small to establish a correlation between uterine perforation and outcomes of subsequent deliveries. Therefore, the statement was modified and now reads “Uterine perforation may be associated to adverse obstetric outcomes.  The possibility of uterine rupture must be considered while managing deliveries of patients after uterine perforation. Moreover, a larger cohort and further studies are needed to establish an association between a uterine perforation and adverse outcomes of subsequent deliveries”.